# Heparan Sulfate Modulation Affects Breast Cancer Cell Adhesion and Transmigration across In Vitro Blood–Brain Barrier

**DOI:** 10.3390/cells13020190

**Published:** 2024-01-19

**Authors:** Yunfei Li, David B. Shteyman, Zeina Hachem, Afaf A. Ulay, Jie Fan, Bingmei M. Fu

**Affiliations:** 1Department of Biomedical Engineering, The City College of the City University of New York, New York, NY 10031, USA; yli012@citymail.cuny.edu (Y.L.); dshteym000@citymail.cuny.edu (D.B.S.); aulay000@citymail.cuny.edu (A.A.U.); 2Department of Natural Sciences, CASL, University of Michigan-Dearborn, Dearborn, MI 48128, USA; hzeina@umich.edu (Z.H.); jiefan@umich.edu (J.F.)

**Keywords:** blood–brain barrier, MDA-MB-231, glycocalyx, vascular endothelial growth factor (VEGF), orosomucoid, sphingosine-1-phosphate (S1P), matrix metalloproteinase (MMP) inhibitor, heparinase III

## Abstract

The disruption of endothelial heparan sulfate (HS) is an early event in tumor cell metastasis across vascular barriers, and the reinforcement of endothelial HS reduces tumor cell adhesion to endothelium. Our recent study showed that while vascular endothelial growth factor (VEGF) greatly reduces HS at an in vitro blood–brain barrier (BBB) formed by human cerebral microvascular endothelial cells (hCMECs), it significantly enhances HS on a breast cancer cell, MDA-MB-231 (MB231). Here, we tested that this differential effect of VEGF on the HS favors MB231 adhesion and transmigration. We also tested if agents that enhance endothelial HS may affect the HS of MB231 and reduce its adhesion and transmigration. To test these hypotheses, we generated an in vitro BBB by culturing hCMECs on either a glass-bottom dish or a Transwell filter. We first quantified the HS of the BBB and MB231 after treatment with VEGF and endothelial HS-enhancing agents and then quantified the adhesion and transmigration of MB231 across the BBB after pretreatment with these agents. Our results demonstrated that the reduced/enhanced BBB HS and enhanced/reduced MB231 HS increase/decrease MB231 adhesion to and transmigration across the BBB. Our findings suggest a therapeutic intervention by targeting the HS-mediated breast cancer brain metastasis.

## 1. Introduction

Tumor metastasis via blood or lymphatic circulations is the leading cause of cancer-related death among cancer patients [1,2,3]. The ability of tumor cells to adhere to and transmigrate through the microvessel wall at the targeted tissue is crucial in tumor metastasis [4]. To search for the effective anti-metastatic therapies, many in vivo and in vitro studies have been conducted to understand underlying mechanisms by which tumor cells interact with endothelial cells lining the microvessel wall for adhesion and transmigration or extravasation [5,6,7,8,9,10,11,12,13].

There is a matrix-like glycocalyx layer at the surface of every mammalian cell including endothelial cells lining the inner wall of our blood vessels and tumor cells [14,15,16,17,18,19,20]. The glycocalyx is comprised of glycoproteins, acidic oligosaccharides, terminal sialic acids, proteoglycans (mainly syndecans and glypicans), and glycosaminoglycans (GAGs) (primarily heparan sulfate, chondroitin sulfate, and hyaluronic acid or hyaluronan) [14,18,21,22,23]. Confined in this glycocalyx layer, there are soluble and insoluble components such as plasma proteins, enzymes, cofactors, and enzyme inhibitors [20,24].

Endothelial glycocalyx is a mechanosensor sensing the blood flow, a regulator controlling the vessel wall permeability, and a barrier restricting the interaction between circulating blood cells and circulating tumor cells with endothelium [19,23,25]. Heparan sulfate (HS) is the most abundant GAG of the endothelial glycocalyx, accounting for 50–90% of all GAGs [22]. By the direct injection of breast cancer cells, MDA-MB-231 (or MB231), into individual post-capillary venules in rat mesentery under physiological flows, Cai et al. [26] found that the HS of the microvessel wall reduced to ~18% of the control after ~45 min of MB231 cell perfusion. By applying heparinase III to pretreat the vessel for 1 h, they found that ~80% of the HS at the microvessel was removed and the MB231 adhesion increased by more than two folds. Similar results were observed in another study investigating MB231 adhesion to and transmigration across an in vitro blood–brain barrier (BBB) formed by bEnd3 (mouse brain microvascular endothelial cells) [27]. It was found that 1 h MB231 adhesion to this in vitro BBB degrades its HS to ~40% of the control. Pretreatment with heparinase III reduced the BBB HS by ~60% and increased MB231 adhesion by 2.5 folds. On the contrary, reinforcing the endothelial glycocalyx of the microvessel wall by a plasma protein, orosomucoid, or a plasma sphingolipid, sphingosine-1-phosphate (S1P), or a generic matrix metalloproteinase (MMP) inhibitor, GM6001, significantly decreased the MB231 adhesion [26,28]. Prior studies also investigated the effect of VEGF (VEGF_165_), vascular endothelial growth factor, a tumor secretion, on microvessel permeability and tumor cell adhesion/transmigration [29,30]. In an in vivo study, Shen et al. [30] showed that 1 h of 1 nM VEGF treatment on a post capillary venule of rat mesentery increased its permeability and enhanced breast cancer cell (MDA-MB-435s) adhesion. In contrast, the pretreatment of the microvessel with SU-1498, an inhibitor to VEGF receptor 2 (VEGFR2, KDR/Flk-1), or pretreatment of the tumor cell with anti-VEGF antibodies not only decreased tumor cell adhesion compared to no pretreatments but also significantly reduced the adhesion enhanced by VEGF. In parallel with the in vivo study, Lee et al. [31] used an in vitro Transwell system with human brain microvascular endothelial cell (HBMEC) monolayer to investigate VEGF effects on tumor cell adhesion and transmigration. They found that VEGF increased MB231 adhesion and transmigration by increasing HBMEC monolayer permeability to inulin. In a more recent study, Fan and Fu [27] used bEnd3 monolayer to investigate the adhesion and transmigration of MB231. They showed that VEGF-enhanced bEnd3 monolayer permeability and MB231 adhesion and transmigration were due to the degradation of the endothelial glycocalyx and the disruption of endothelial junction proteins [27,32,33,34]. Mensah et al. [35] also found that a reduced glycocalyx due to a disturbed flow induced the entry of circulating tumor cells into the endothelium.

As presented above, the glycocalyx of endothelial cells prevents tumor cell adhesion and transmigration. In contrast, recent studies found that glycocalyx at tumor cells promotes tumor metastasis [18,36,37]. Compared to healthy cells, cancer cells have a thicker and denser glycocalyx [18,38,39,40]. The glycosaminoglycan content isolated from tissue containing lethal breast cancer tumors was approximately twice that of other tissues [41]. Recent studies also observed that a bulkier glycocalyx on tumor cells is associated with an increased migration and the metastatic potential of cancers [42]. The tumor cell glycocalyx responds to the interstitial flow-induced shear forces by secreting matrix metalloproteinases (MMPs) to degrade the ECM of surrounding tissues. This makes it easier for tumor cells to migrate through the tissue and invade the nearby vasculature [43]. The glycocalyx also determines whether tumor cells can migrate against the direction of shear flow to reach the vasculature [44,45]. The high sialic acid content of the glycocalyx on the circulating tumor cells helps them to escape from the immune surveillance [46,47]. The bulky glycocalyx on the circulating tumor cells, such as hyaluronic acid, not only creates a barrier to therapeutic agents, but also a shield to the blood flow-induced friction forces [48].

In summary, prior studies found that glycocalyx of tumor cells promotes tumor cell migration in the extracellular space, and a tumor cell secretion, VEGF, disrupts the endothelial glycocalyx to increase tumor cell adhesion/transmigration. Our most recent investigation employing super-resolution stochastic optical reconstruction microscopy (STORM) revealed that there was a differential effect of VEGF on the glycocalyx of endothelial and tumor cells [49]. We found that while VEGF significantly reduced the length and coverage of heparan sulfate (HS) on hCMEC (human cerebral microvascular endothelial cell) monolayer, an in vitro blood–brain barrier (BBB) model, it did not change the thickness and coverage of hyaluronic acid (HA) on the BBB. On the contrary, VEGF significantly enhanced the coverage of HS and HA on MB231 although it did not alter their thickness.

However, how the enhanced HS of MB231 and the reduced HS of the BBB due to VEGF or the modulated HS of the glycocalyx of endothelial and tumor cells can influence tumor cell adhesion/transmigration to/across endothelial barriers is largely unknown. Therefore, in this study, we tested the hypothesis that reducing/enhancing breast cancer cell (MB231) HS while enhancing/reducing HS at the BBB can inhibit/promote MB231 adhesion/transmigration to/across the BBB. To test this hypothesis, we generated an in vitro BBB by culturing hCMECs on either a glass-bottom dish or a Transwell filter. We first quantified the HS of the BBB and that of MB231 after treatment with VEGF, or with endothelial HS-enhancing agents, orosomucoid, sphingosine-1-phosphate (S1P), and a generic matrix metalloproteinase (MMP) inhibitor, GM6001, or with an enzyme, heparinase III, which can degrade endothelial HS. Then, we quantified the adhesion and transmigration of MB231 to/across the BBB after pretreatment with these agents to (1) MB231 only, (2) BBB only, and (3) both MB 231 and BBB.

## 2. Materials and Methods

### 2.1. Cell Culture

Human cerebral microvascular endothelial cells (hCMEC/D3 or hCMEC) from Millipore Sigma (Burlington, MA, USA) (passage 7 to 20 after purchase) were cultured using EBM^TM^-2 Basal Medium (Lonza, Basel, Switzerland), supplemented with EGM^TM^-2 MV Microvascular Endothelial Cell Growth Medium SingleQuots^TM^ kit (Lonza) [49]. Human breast carcinoma cells (MDA-MB-231 or MB231) from ATCC (Manassas, VA, USA) (passage 10 to 18 after purchase) were cultured in Dulbecco’s Modified Eagle’s Medium/Nutrient Mixture F-12 Ham (DMEM/F-12), 2 mM L-glutamine, 100 U/mL penicillin, and 1 mg/mL streptomycin, all from Sigma-Aldrich (St. Louis, MO, USA), supplemented with 10% fetal bovine serum (FBS, Atlanta Biologicals, Flowery Branch, GA, USA) [26,27,28]. Human nontumorigenic breast epithelial cell line MCF-10A cells (ATCC) were cultured in MEGM bullet kit (Lonza) supplemented with 100 ng/mL cholera toxin (Sigma-Aldrich) as described in [26]. All cells were cultured in the incubator with 5% CO_2_ at 37 °C.

### 2.2. Generation of In Vitro BBBs

We generated in vitro BBBs by culturing endothelial monolayers either on a glass-bottom dish or on a Transwell filter. No. 1.5 glass-bottom dishes (MetTek, Ashland, MA, USA) or Transwell filters (VWR, Radnor, PA, USA) with an 8 μm pore transparent PET membrane (0.33 cm^2^ bottom area), and they were first coated with 50 μg/mL Collagen I (Sigma-Aldrich) and incubated overnight in an incubator at 37 °C. Then, hCMECs were seeded at a density of 60 k/cm^2^ on a glass-bottom dish or a Transwell filter and cultured for ~5 days until confluent as in our previous studies [27,49]. The formation of an in vitro BBB was determined by the transendothelial electrical resistance (TEER) of the monolayer, which was measured by a chopstick-shaped Volt/Ohm meter (EVOM2™, World Precision Instruments, Sarasota, FL, USA). The in vitro BBB was considered to be generated when the TEER of the monolayer was unchanged in two consecutive days. The TEER of a blank Transwell filter with the same cell culture medium was also measured and subtracted from the TEER of the total system to determine the TEER of the in vitro BBB. Figure A1 demonstrates a schematic for measuring the TEER of the in vitro BBB generated on a Transwell filter.

### 2.3. Quantification of Heparan Sulfate (HS) at In Vitro BBB and MB231

For the quantification of HS at MB231, the MB231 cells were first seeded at a density of 20 k/cm^2^ on a 30 μg/mL fibronectin-coated No. 1.5 glass-bottom dish and adhered for 0.5–1 h. The non-adherent cells were washed away before HS labeling. The HS of the in vitro BBB generated in the above section and the adherent MB231 were then labeled by immunostaining. Samples were first rinsed by 10 mg/mL bovine serum albumin (BSA, Sigma-Aldrich) in PBS (1%BSA/PBS), then fixed with 2% paraformaldehyde and 0.1% glutaraldehyde for 20 min, and 0.1% NaBH_4_ (Sigma-Aldrich) for 7 min. After washing 3× with 1% BSA/PBS, the samples were blocked by 2% normal goat serum for 30 min at room temperature (RT). Then, the samples were incubated with an anti-heparan sulfate antibody (1:100, 1 × 10^4^ epitope, mouse, Amsbio, Abingdon, UK) at 4 °C overnight, followed by an Alexa Fluor 488 conjugated goat anti-mouse IgG (Alexa Fluor™ 488, 1:200; Invitrogen, Thermo Fisher Scientific, Waltham, MA, USA) for 1 h at RT [27,49]. Finally, the samples were mounted using FluoromountG (SouthernBiotech™, Birmingham, AL, USA) with DAPI after a DPBS wash. For the samples on the Transwell filter, the membrane of the filter was cut and made into slides for confocal imaging.

The samples were scanned using a Zeiss LSM 800 confocal laser scanning microscope with a 40×/NA1.3 oil immersion objective lens. Five fields (each field of 320 µm × 320 µm) (2048 × 2048) from each sample were captured as a z-stack of 30–40 images with a z-step of 0.32 μm. Image projection and intensity quantification for HS were performed with Zeiss ZEN and NIH ImageJ 1.53t [27]. The HS at the BBB with adherent MB231 cells was also quantified using this protocol but with more images collected.

### 2.4. Modulation of HS of In Vitro BBB and MB231 by Various Agents

Zeng et al. [50] applied different concentrations of heparinase III for 2 h to digest HS on the rat fat pad endothelial cell monolayer, and Cai et al. [26] perfused 1% BSA Ringer solution with 50 mU/mL heparinase III into a rat mesenteric postcapillary venule for 1 h to disrupt HS at the luminal surface of a microvessel in vivo. Based on their studies, we applied 50 mU/mL heparinase III (Sigma-Aldrich) for 2 h to manipulate the HS at the in vitro BBB and MB231 cells before investigating MB231 adhesion to and transmigration across the BBB. Xia et al. [49] applied 1 nM VEGF for 2 h on in vitro BBB and MB231 to demonstrate a differential effect on their respective HS. In a study for tumor cell adhesion to the wall of a microvessel on rat mesentery, Shen et al. [30] pretreated the microvessel with 1 nM VEGF for 1 h, or 50 μM SU-1498 (an inhibitor to VEGFR2 (KDR/Flk-1)) for 45 min, and pretreated the tumor cells with 20 μg/mL anti-human VEGF monoclonal antibody for 1 h. They showed that VEGF increased tumor cell adhesion while anti-VEGF and SU-1498 reduced the adhesion. Corresponding to these studies, we pretreated the BBB and MB231 with 1 nM VEGF (recombinant human VEGF_165_, Peprotech, Rocky Hill, NJ, USA) for 2 h, and pretreated the BBB with 50 μM SU-1498 (Alomone labs, Ltd., Jerusalem, Israel) for 1.5 h, both at 37 °C, while we pretreated MB231 with 20 μg/mL anti-VEGF (Thermo Fisher Scientific) for 2 h at 4 °C [30]. Cai et al. [26] enhanced the HS of the microvessel wall to 1.4 folds of the control by perfusing 0.1 mg/mL orosomucoid in 1% BSA-Ringer into a rat mesenteric postcapillary venule for 30 min. Zhang et al. [28] also increased microvessel HS by the pretreatment of the microvessel with 1 μM sphingosine-1-phosphate (S1P) or 10 μM GM 6001 (a generic MMP inhibitor) for 20 min. Both found that the enhanced HS at the microvessel by orosomucoid, S1P, or GM 6001 decreased MB231 adhesion to the microvessel. Therefore, in our study, we pretreated in vitro BBB and MB231 for 1 h with 0.1 mg/mL orosomucoid (G3643, α1-acid glycoprotein from bovine plasma, Sigma-Aldrich) or 1 μM S1P (73914; Sigma-Aldrich) or 10 μM GM 6001 and its negative control GM 6001 NC (Sigma-Aldrich). To make the stock solutions, S1P was dissolved in 95% methanol, while GM 6001 and GM 6001 NC were dissolved in DMSO, as described in Zhang et al. [28].

### 2.5. Determination of Solute Permeability (P) of In Vitro BBB

We employed the same protocol in [27] to determine the solute permeability (P) of the in vitro BBB generated on a Transwell filter of 0.9 cm^2^ surface area. Briefly, the upper chamber of the filter was loaded with 0.5 mL of 10 μM FITC–Dex70k (Dextran-70k, MW 70 kD, Sigma-Aldrich) in 10 mg/mL BSA in a Ringer solution. Then, 1.5 mL of the same solution without FITC-Dex 70k was added to the lower chamber. The 50 μL solution in the lower chamber was taken out every 10 min for 90 min, and the lower chamber was refilled with 50 μL BSA-Ringer solution. The intensity of the sample solution with FITC-Dex70k was measured by using a SpectraMax M5 microplate reader (Molecule Devices, San Jose, CA, USA). The permeability $\left(P_{m}\right)$ to Dex-70k was calculated by $P_{m}=\frac{∆I_{L}/∆t\times V}{I_{U}\times A}$, where $\frac{∆I_{L}}{∆t}$ is the increased rate of the fluorescent intensity of the solution in the lower chamber during the time interval ∆*t*, $I_{U}$ is the fluorescence intensity in the upper chamber, *V* is the solution volume in the lower chamber, and A is the area of the Transwell filter membrane. Calibration experiments for the concentration vs. intensity for the FITC-Dex70k were performed to ensure that the concentration was linearly related to the intensity of the solution used in our study. The permeability of the in vitro BBB ($P_{BBB})\;$ to Dex-70k was calculated by using $1/P_{m}=1/P_{b}+1/P_{BBB}$, where $P_{m}$ is the measured permeability of both the BBB and the Transwell filter, $P_{b}$ is the measured permeability of the blank transwell filter, and $P_{BBB}$ is the permeability of the in vitro BBB. Figure A1 demonstrates a schematic for determining the solute permeability of the in vitro BBB generated on a Transwell filter. The detailed protocol is described in [51].

### 2.6. Quantification of MB231 Cell Adhesion to the In Vitro BBB Generated on a Glass-Bottom Dish

MB231 cells were first fluorescently labeled with 10 μM cell tracker red, EX/EM = 577/602 nm (Invitrogen, Thermo Fisher Scientific), for 30 min, trypsinized, washed with 1% BSA-Ringer, and filtered using a 40 µm cell strainer [26,28]. Under no treatment, or pretreatment with various agents described in the previous section, the labeled MB231 cells in 1% BSA-Ringer were placed on the in vitro BBB at a glass-bottom dish at a density 60 k/cm^2^ and incubated for 2 h. After gently washing away the non-adherent MB231 cells, the BBB with adherent MB231 cells were imaged using a Nikon Eclipse TE2000-S microscope with an objective lens 20×/NA0.75 [33]. Overall, 3–4 fields (334 μm × 436 μm for each field) were imaged for each sample with at least 10 fields from three independent experiments analyzed for each case. The number of adherent MB231 was counted for each case and normalized by the averaged number from the case with no treatment. Finally, the in vitro BBB and adherent MB231 were fixed with 2% paraformaldehyde and 0.1% glutaraldehyde and used for HS immunostaining.

### 2.7. Quantification of MB231 Cell Transmigration across the In Vitro BBB Generated on a Transwell Filter

We quantified the transmigrated MB231 cells following the protocol described in [27]. Under no treatment, or pretreatment with various agents described in the previous section, the fluorescently labeled MB231 cells in 1% BSA-Ringer were placed on the in vitro BBB on a Transwell filter at a density 60 k/cm^2^ and incubated for 6 h. After removing non-adherent cells, the BBB along with the adherent, transmigrating, and transmigrated MB231 cells were fixed with 2% paraformaldehyde and 0.1% glutaraldehyde and immunostained with anti-HS and fluorescently labeled antibody as described in the previous section. Finally, the samples were mounted using Fluoromount-G (SouthernBiotech™) with DAPI after DPBS wash and made into slides for confocal imaging. The samples were scanned by a Zeiss LSM 800 confocal laser scanning microscope with a 40×/NA1.3 oil immersion objective lens. Five fields (each field 320 µm × 320 µm) (1024 × 1024) from each sample were captured as a z-stack of 60–80 images with a z-step of 0.59 μm. Four samples from four independent experiments were analyzed for each case. The number of transmigrated MB231 cells was counted for each case and normalized by the averaged number from the case with no treatment.

The viability rate was >95% for both hCMECs and MB231 cells under all the treatments and after 6 h incubation.

### 2.8. Statistical Analysis

For all the measurements, data were presented as mean ± standard deviation (SD). A *t*-test or two-way ANOVA was used for comparisons between treatments and non-treatment and among different treatments. The samples were obtained from at least three independent experiments. A level of *p*-value < 0.05 was considered statistically significant in all the experiments.

## 3. Results

### 3.1. Differential Effects of VEGF, Orosomucoid, and S1P on the HS of In Vitro BBB and MB231

We first manipulated HS at the surface of the in vitro BBB and MB231 by various agents as described in the earlier section. Figure 1 shows the confocal images and intensity quantification of the HS at the BBB under no treatment (control) and after the treatment with various agents. Correspondingly, Figure 2 shows those for MB231. As expected, heparinase III reduces the HS at the BBB and MB231 to 63% and 50% of the controls, respectively. Also, same as in Xia et al. [49], VEGF decreases the HS of the BBB to 29% of the control while increases that of MB231 to 1.5 folds of the control. We also compared the HS of benign breast epithelial cells MCF-10A with that of MB231 cells. MCF-10A cells have much less HS, only ~20% that of MB231 cells. In addition, VEGF seems not to increase HS of MCF-10A (Figure A2). Although orosomucoid and S1P enhance the HS of the BBB to 3.4 and 1.9 folds of the control, respectively, like their effects on the HS of the microvessels in vivo [26], surprisingly, they reduce the HS of MB231 to 64% and 73% of the control, correspondingly. Similar to their effects on the HS of the microvessels in vivo, a generic MMP inhibitor GM 6001 enhances the HS of the in vitro BBB to 2.2 folds of the control while SU-1498, an inhibitor to VEGFR2, does not alter the HS of the BBB. In contrast, GM 6001 has no effect on the HS of MB231 or on anti-VEGF.

### 3.2. Effects of HS Modulation on the Barrier Functions of In Vitro BBB

Because the endothelial surface glycocalyx is a major barrier of the BBB to macromolecules [52] and HS is the most abundant component of the glycosaminoglycans of the endothelial glycocalyx [22,23,53], we first examined the changes in the BBB permeability due to Dextran-70k (P^Dex−70k^) after HS modulation by various agents. Figure 3A shows that the disruption of HS by heparinase III and VEGF increases P^Dex−70k^ to 1.6 and 1.5 folds of the control, respectively. In contrast, reinforcing HS by orosomucoid, S1P, and GM 6001 decreases P^Dex−70k^ to 71%, 81%, and 63% of the control, correspondingly. Previously, it was found that VEGF not only degrades the HS but also disrupts the endothelial tight junctions in vitro [33] and in vivo [34]. As tight junctions are the major barrier to ions and small molecules, we also examined the effects of various HS-modulating agents on the TEER of the BBB since the reciprocal of the TEER represents the permeability to ions [51,52]. Figure 3B demonstrates that endothelial HS-disrupting agents, heparinase III and VEGF, also slightly disrupts endothelial tight junctions by decreasing the TEER by 9% and 14% than that of the control, respectively. Endothelial HS-enhancing agents orosomucoid and S1P moderately increase the TEER by 11% and 13% than that of the controls, correspondingly, suggesting they also reinforce the endothelial junctions. Although HS-enhancing agent GM 6001 decreases the BBB permeability to dextran-70k, it does not change the TEER, suggesting that GM 6001 only affects the glycocalyx. The inhibition of VEGFR2 by SU-1498 has no effect on either the BBB permeability to dextran-70k or the TEER.

### 3.3. Effects of HS Modulation on MB231 Adhesion to and Transmigration across In Vitro BBB

To test the effects of HS modulation on MB231 adhesion to and transmigration across the in vitro BBB, we performed three types of modulation: pretreatments with various agents to MB231 only, to in vitro BBB only, and to both MB231 and the BBB. Figure 4 shows the effects of HS modulation on MB231 adhesion to the in vitro BBB generated on the glass-bottom dish. For pretreatments to MB231 only, the degradation of HS at MB231 by heparinase III, orosomucoid, and S1P reduces MB231 adhesion to 68%, 83%, and 77%, respectively, compared to that with no treatment. On the contrary, the enhanced HS at MB231 by VEGF significantly increases the MB231 adhesion to 1.4 folds of that with no treatment (Figure 4A). For pretreatments to the BBB only, the decreased HS at the BBB by heparinase III and VEGF increases the MB231 adhesion to 1.3 and 1.7 folds of that with no treatment, while the reinforced HS at the BBB by orosomucoid, S1P, and GM 6001 reduces the MB231 adhesion to 69%, 68%, and 73%, respectively, compared to that with no treatment (Figure 4B). For pretreatments to both the BBB and MB231, the reduced HS at both MB231 and the BBB by heparinase III does not alter the MB231 adhesion compared to that with no treatment, while the enhanced HS at MB231 but the reduced HS at the BBB by VEGF further increases the MB231 adhesion to 2.3 folds compared to that with no treatment, causing significantly more adhesion than that with pretreatment to either MB231 only or the BBB only. Pretreatments with orosomucoid, S1P, and GM 6001 to both the BBB and MB231 all reduce the MB231 adhesion, but no significant difference is observed in the reduction compared to that with pretreatments to the BBB only, or that to MB231 only (except pretreatment with GM 6001) (Figure 4C). Figure 4D demonstrates typical fluorescent microscopic images for adherent MB231 cells to the BBB after various pretreatments for both MB231 cells and the BBB.

Figure 5 shows the effects of HS modulation on MB231 transmigration across the in vitro BBB generated on the Transwell filter. For pretreatments to MB231 only, the degradation of HS at MB231 by heparinase III, orosomucoid, and S1P reduces MB231 transmigration to 26%, 58%, and 68%, respectively, compared to that with no treatment. This is consistent with their effects on the adhesion except that the degradation of HS at MB231 by heparinase III has a larger effect on the transmigration than on the adhesion. The VEGF-enhanced HS at MB231 increases MB231 transmigration to 1.5 folds of that with no treatment, comparable to the increase in the adhesion (Figure 5B). For pretreatments to the BBB only, the decreased HS at the BBB by heparinase III and VEGF increases the MB231 transmigration to 1.9 and 1.5 folds of that with no treatment, while reinforced HS at the BBB by orosomucoid, S1P, and GM 6001 reduces the MB231 transmigration to 55%, 64%, and 39%, respectively, compared to that with no treatment (Figure 5C). It seems that the modulation of HS at the BBB by these agents has a larger effect on the MB231 transmigration than on its adhesion except VEGF. For pretreatments to both the BBB and MB231, the reduced HS at both MB231 and the BBB by heparinase III does not alter the MB231 transmigration compared to that with no treatments, and this is the same effect as for the adhesion. Consistent with its effect on the adhesion, the enhanced HS at MB231 and the reduced HS at the BBB by VEGF increases the MB231 transmigration to 1.8 folds compared to that with no treatment, more increase than that with pretreatment to either MB231 only or the BBB only. Finally, pretreatments with orosomucoid, S1P, and GM 6001 to both the BBB and MB231 all reduce the MB231 transmigration with GM 6001 reducing the most, but no significant difference is observed in the reduction compared to that with pretreatments to the BBB only or that to MB231 only (except pretreatment with GM 6001) (Figure 5D).

We also performed the adhesion and transmigration of benign mammary epithelial cells MCF-10A under no treatment and pretreatment with VEGF and orosomucoid. MCF-10A has only ~20% HS compared to that in MB231, and neither VEGF nor orosomucoid significantly alters its HS (Figure A2). Under no treatment, the number of adherent MCF-10A cells to the BBB is ~1/3 that of adherent MB231 cells to the BBB, and no transmigrated MCF-10A across the BBB was observed after 6 h incubation. Neither pretreatment with VEGF nor orosomucoid modulated MCF-10A adhesion and transmigration compared to no treatment.

### 3.4. Effects of MB231 Adhesion and Various Pretreatments on HS of In Vitro BBB

To investigate if MB231 adhesion further modulates the HS of the in vitro BBB in addition to the pretreatments of various agents, we quantified the HS of the BBB in a region of interest (ROI) with adherent MB231 cells and that in a ROI without adherent MB231 cells (see the definitions for the ROIs with and without adherent MB231 cells in the caption of Figure 6). Figure 6 demonstrates our findings for after 2 h MB231 adhesion to the BBB generated on the glass-bottom dish. Without pretreatment to either MB231 or the BBB, the HS in the ROI with adherent MB231 is 55–83%, with an average of 67% HS compared to that of the ROI without adherent MB231. Two-hour adhesion of MB231 indeed degrades more HS of the BBB at adhesion sites for the case without any pretreatment. Interestingly, for all the pretreatments except VEGF, the HS of the BBB in the ROI with and that without adherent MB231 are not different from each other. Pretreatment with heparinase III to MB231 degrades its HS and reduces its adhesion to the BBB and its ability to further degrade the HS of the BBB at the adhesion sites (Figure 6A). On the contrary, pretreatment with VEGF to MB231 enhances the HS of MB231 and increases its adhesion to the BBB and its ability to further disrupt the HS of the BBB at the adhesion site. In addition, pretreatment with VEGF to the BBB degrades its HS and increases MB231 adhesion, which disrupts more the HS of the BBB at the adhesion site. Although there is a very minor difference between the HS of the ROI with adherent MB231 under no pretreatment and that with pretreatment with VEGF to MB231, or to the BBB, or to both, there is no difference in the HS of the ROI with adherent MB231 among various pretreatments with VEGF (Figure 6B). These findings suggest that MB231 adhesion to the BBB is more disruptive to the HS of the BBB than VEGF. Reducing the HS at MB231 while reinforcing the HS at the BBB using orosomucoid, S1P, and GM 6001 not only decrease MB231 adhesion to the BBB but also protect the HS of the BBB at the adhesion site (Figure 6C–E).

## 4. Discussion

Prior studies have shown that the disruption of HS at endothelial cells by VEGF or heparinase III promotes breast cancer cell adhesion to the microvessel wall and transmigration across an in vitro BBB [26,27,30]. The results from our current study on the effect of cancer cell adhesion/transmigration by modulating endothelial HS are consistent with these previous studies. It has also been indicated that tumor glycocalyx participates in cancer cell progression and metastasis [36] by affecting transmembrane receptor function, cellular tension, integrin-mediated signaling, cell–cell and cell–ECM interactions, and immune recognition [18,43,46,47,48]. It was found that 95% of breast cancer cells have a modified glycocalyx composition or structure that also reshapes their function compared to the glycocalyx of a healthy cell [54]. Tumor cells are characterized by a thicker and higher density glycocalyx. High glycoprotein levels are abundantly expressed in circulating tumor cells. The thick glycocalyx of tumor cells promotes metastasis even on soft substrate surfaces by mechanically enhancing cell surface receptor function [38]. Consistent with these previous observations, our results showed that malignant breast cancer cell MB231 has much higher HS density, ~5 folds that of benign cell MCF-10A, which increases MB231 adhesion to an in vitro BBB by ~3 folds compared to MCF-10A adhesion.

However, whether modulating tumor glycocalyx, particularly HS, would affect tumor cell adhesion/transmigration to/across endothelium for hematogenous metastasis is unclear. Our results demonstrated that VEGF_165_, a tumor secretion, especially overexpressed by breast cancer cell MB231 [55], greatly disrupts the HS on the BBB and increases the BBB permeability to a large molecule, Dex-70K, and to ions (decreasing TEER), but it enhances the HS of MB231 by 1.5 folds after 2 h treatment with 1 nM VEGF_165_. The differential effect of VEGF on the HS of MB231 and the BBB is consistent with that observed by Xia et al. [49]. Interestingly, the enhanced HS of MB231 by VEGF increases MB231 adhesion to and transmigration across the in vitro BBB by 1.4 and 1.5 folds, respectively, if VEGF pretreatment is only on MB231; it further increases the adhesion and transmigration by 2.3 and 1.8 folds, correspondingly, if VEGF pretreatment is on both MB231 and the BBB. On the other hand, reducing the HS of MB231 by heparinase III decreases MB231 adhesion and transmigration, while reducing the HS of the BBB by heparinase III increases MB231 adhesion. However, reducing HS of both MB231 and the BBB does not change the adhesion/transmigration compared to no HS alteration.

It is understandable that the degradation of HS on the BBB by VEGF or by heparinase III enables more VEGF receptors and ligands of cell adhesion molecules (CAMs) at MB231 to interact with VEGF and CAMs at MB231 to enhance their adhesion. In addition, VEGF and heparinase III disrupt adherens and tight junctions between endothelial cells [33,34] to expose more ECM proteins, such as laminins and fibronectins [33,56], which interact with CAMs at MB231 cells, e.g., integrins [18,33], to promote their adhesion and transmigration. Prior studies found that either SU-1498, an inhibitor to VEGFR2 at endothelium, or anti-VEGF antibody to tumor cells can inhibit tumor cell adhesion to endothelium [30,33], and our results showed the same inhibition effects of SU-1498 and anti-VEGF although neither of them alter the HS of the BBB or that of MB231, suggesting that their inhibition abilities are independent of HS modulation.

Nevertheless, it is quite puzzling why the enhanced HS of MB231 by VEGF increases MB231 adhesion and transmigration. One possibility is that the HS-associated heparan sulfate proteoglycans (HSPGs) of MB231 can bind to CAMs of endothelial cells (the BBB) and ECM proteins for adhesion and transmigration. HSPGs are the common constituents of cell surfaces and the ECM. HSPGs interact with many proteins including growth factors, chemokines, and structural proteins of the ECM to influence cell growth, differentiation, and the cellular response to the environment [57,58]. HSPGs comprise a protein core to which the chains of glycosaminoglycan (GAG) and heparan sulfate (HS) are covalently attached during post-translational modification. At the cell surface, the two major families of HSPG are the transmembrane syndecans and the GPI-anchored glypicans [58,59]. HSPG promoted MB231 cells spreading, focal adhesion, and adherens junction formation [60]. HSPGs on the melanoma cells were found to interact with p-selectins on endothelial surfaces for adhesion under flow [61]. The overexpression of syndecan-1 in highly metastatic murine lung carcinoma cells enhances pulmonary metastasis when cells are injected intravenously [62]. The HS of tumor cells was shown to participate in tumor cell adhesion to ECM molecules including fibronectin, laminins, and collagens [56]. By using super-resolution STORM, Xia et al. [49] revealed that VEGF significantly enhanced the coverage of HS on MB231 although it did not alter its thickness. The enhanced coverage of HS is associated with more HSPGs, which bind to more CAMs of the BBB and ECM proteins, resulting in increased adhesion and transmigration.

Li et al. [63] employed AFM (atomic force microscopy) to measure the mechanical properties of breast cancer (MCF-7) and benign (MCF-10A) cells. They found that malignant cells had an apparent Young’s modulus which was 1.4–1.8 times lower than that of benign cells. AFM and confocal microscopy further revealed reduced and disorganized actin filaments in malignant cells, which is one contributing factor for them being softer than benign cells. A softer cell cytoskeleton with disrupted actin filaments enhances cancer cell invasion and migration but reduces its defense ability to the surrounding ECM and blood flow-induced disruption forces. Thus, more glycocalyx coating is necessary for cancer cells. To investigate the morphological changes in MB231 during adhesion and transmigration, we compared the ratio of the cell height to length after 2 h and 6 h incubation with the BBB. Figure 7 demonstrates that MB231 becomes much flatter after longer time adhesion. The flatter or more spreading MB231 surface enables more HS or HSPGs to bind to the CAMs of the BBB and ECM proteins, promoting MB231 adhesion and transmigration.

Orosomucoid is a plasma protein essential for the maintenance of stable microvessel solute permeability by enhancing the charge and organization of the endothelial glycocalyx [64,65]. It has been reported that the orosomucoid level increases several-fold during infection or trauma [66,67] in order to reduce the transvascular leakage of albumin [68]. Cai et al. [26] showed that 30 min of 0.1 mg/mL orosomucoid treatment increased the HS of the post-capillary venule to 1.4 folds on rat mesentery compared to that of no treatment. Consequently, it reduced MB231 adhesion by ~46% in 60 min. Our current results also demonstrated that 1 h treatment of the BBB with 0.1 mg/mL orosomucoid enhanced the HS of the BBB by 3.4 folds and reduced MB231 adhesion and transmigration by ~31% and ~45%, respectively. On the other hand, orosomucoid surprisingly degraded the HS at MB231 by ~36% and decreased its adhesion and transmigration by ~17% and ~42%, respectively.

Similar results as for orosomucoid in modulating the HS of the BBB and MB231 were observed with the treatment of sphingosine-1-phosphate (S1P), a sphingolipid in plasma that plays a critical role in the cardiovascular and immune systems [69]. Red blood cells (RBCs) are a major source of S1P in plasma, which acts continuously to maintain normal vascular permeability under physiological conditions [70,71,72]. Zhang et al. [28] demonstrated that the treatment with S1P enhanced the HS of the rat mesenteric post-capillary venule and reduced MB231 adhesion. The possible mechanism by which S1P enhances the endothelial HS is that S1P activates S1P receptor 1 (S1PR1). The activation of S1PR1 inhibits the activity of MMPs and abolishes MMP activity-dependent syndecan-1 ectodomain shedding, which induces the loss of attached HS [73]. Thus, GM6001, an inhibitor to MMPs, produces the same effects as S1P in preserving the HS of the BBB and reducing MB231 adhesion and transmigration. The activation of S1PR1 also increases myosin light chain phosphorylation that tightens the junctions between the endothelial cells forming the microvessel wall, which inhibits lung metastasis by up to 80% in a melanoma metastasis animal model [74]. We also found that S1P enhanced the TEER of the BBB, indicating reinforced endothelial junctions as observed in Chen et al. [74]. On the other hand, GM6001 did not alter the TEER, indicating that GM6001 only protects glycocalyx by inhibiting MMP activity.

But how orosomucoid and S1P reduce the HS of MB231 remains to be elucidated. However, their differential effects on the HS of MB231 and the BBB can diminish MB231 adhesion to and transmigration across the BBB.

The current study utilizing an in vitro BBB model, in which human cerebral microvascular endothelial cells were used, is still necessary to be validated in an animal model. We previously showed that reducing/enhancing HS at the microvessel of rat mesentery increased/decreased MB231 adhesion under physiological conditions. We will validate our current in vitro results in rat cerebral microvessels in vivo in a future study.

## 5. Conclusions

Our results demonstrated that the reduced/enhanced HS of the BBB and the enhanced/reduced HS of MB231 increase/decrease MB231 adhesion to and transmigration across the BBB. MB231 adhesion to the BBB is disruptive to the HS of the BBB. Reducing the HS at MB231 while reinforcing the HS at the BBB using orosomucoid, S1P, and GM 6001 not only decrease MB231 adhesion but also protect the HS of the BBB at the adhesion site. Our findings thus suggest a therapeutic intervention by targeting the HS-mediated breast cancer brain metastasis.

## Figures and Tables

**Figure 1 cells-13-00190-f001:**
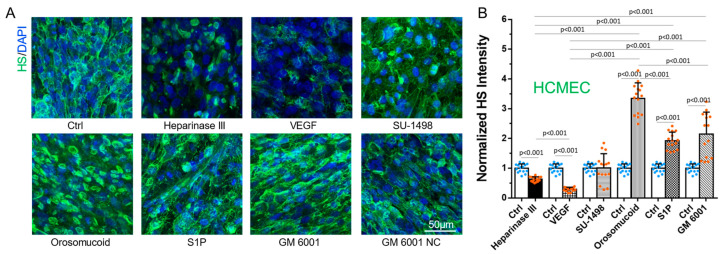
Modulation of heparan sulfate (HS) at an in vitro BBB by various agents. (**A**) Confocal images showing HS (green) at the BBB under control (no treatment) and after the treatment by various agents. (**B**) Normalized HS intensity at the BBB under control (Ctrl) and various treatments. Values are mean ± SD. *n* = 3 samples with 5 fields (each field 320 µm × 320 µm) per sample analyzed for each case. No significant difference between the control and GM 6001 NC (not shown in the plot (**B**)).

**Figure 2 cells-13-00190-f002:**
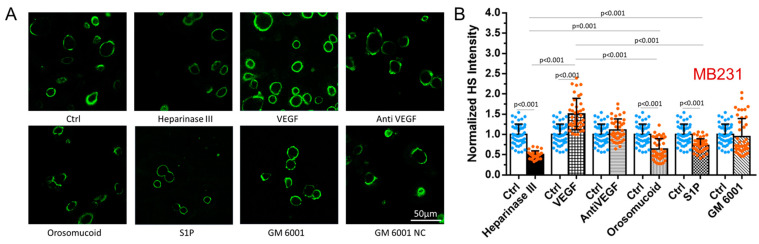
Modulation of heparan sulfate (HS) at MB231 by various agents. (**A**) Confocal images showing HS (green) at MB231 under control (no treatment) and after the treatment by various agents. (**B**) Normalized HS intensity at MB231 under control (Ctrl) and after various treatments. Values are presented as mean ± SD. *n* ≥ 30 cells from 3 independent experiments analyzed for each case. No significant difference between the control and GM 6001 NC (not shown in the plot (**B**)).

**Figure 3 cells-13-00190-f003:**
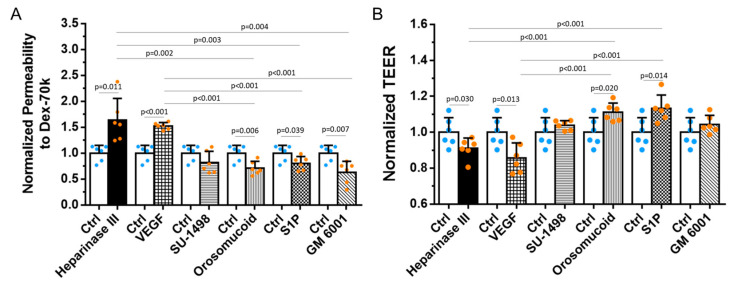
Effects of heparan sulfate (HS) modulation by various agents on the in vitro BBB permeability to dextran-70k and its TEER. (**A**) Normalized permeability to dextran-70k. (**B**) Normalized TEER. Values are presented as mean ± SD. *n* = 6 samples from 3 independent experiments for each case. No significant difference between the control and GM 6001 NC (not shown in the plots).

**Figure 4 cells-13-00190-f004:**
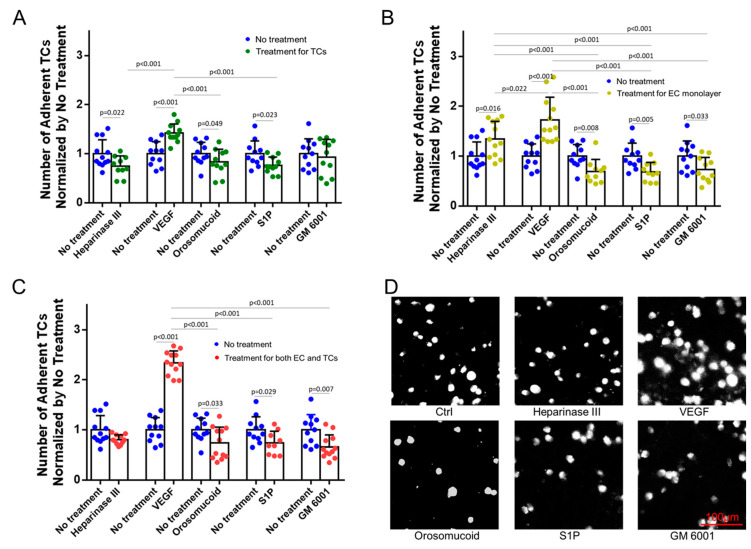
Effects of heparan sulfate (HS) modulation by various agents on MB231 adhesion to in vitro BBB. Number of adherent MB231 cells to the BBB under no treatment (control) and after various pretreatments: (**A**) for MB231 cells only, (**B**) for the BBB (EC monolayer) only, and (**C**) for both MB231 cells and the BBB. (**D**) Fluorescent microscopic images for adherent MB231 cells to the BBB after various treatments for both MB231 cells and the BBB. Values are presented as mean ± SD. *n* ≥ 10 fields (334 µm × 436 µm for each field) from 3 independent experiments analyzed for each case. No significant difference between no treatment and GM 6001 NC (not shown in the plots).

**Figure 5 cells-13-00190-f005:**
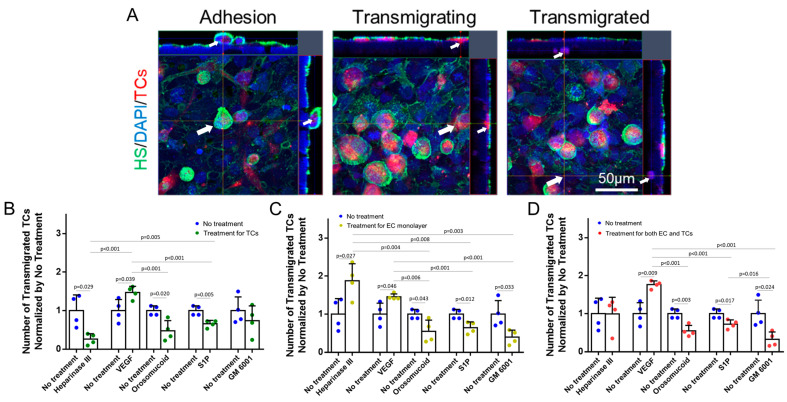
Effects of heparan sulfate (HS) modulation by various agents on MB231 transmigration across an in vitro BBB. (**A**) Confocal images for adherent, transmigrating, and transmigrated MB231 cells to/across the BBB. Arrows indicate the corresponding cells in each state. Number of transmigrated MB231 cells across the BBB under no treatment (control) and after various pretreatments (**B**) for MB231 cells only, (**C**) for the BBB (EC monolayer) only, and (**D**) for both MB231 cells and the BBB. Values are presented as mean ± SD. *n* = 4 independent experiments analyzed for each case. No significant difference between no treatment and GM 6001 NC (not shown in the plots).

**Figure 6 cells-13-00190-f006:**
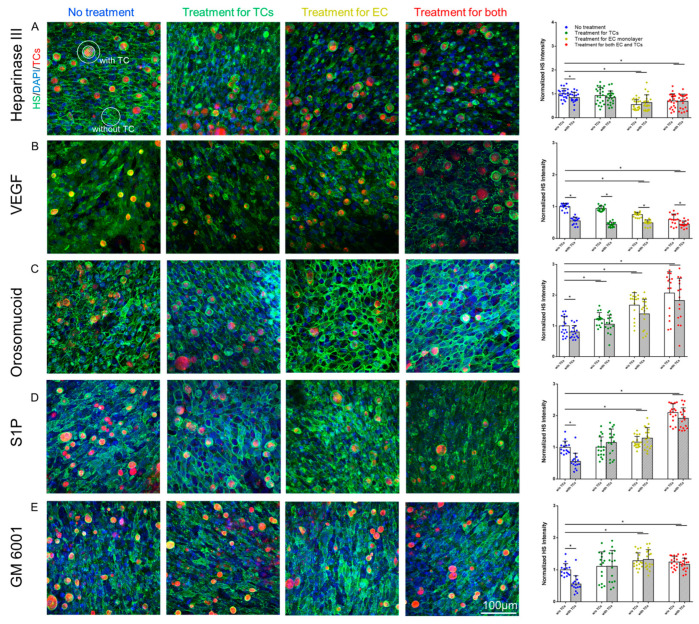
Effects of MB231 adhesion and various treatments on heparan sulfate (HS) at an in vitro BBB. The left panels show the confocal images of HS at the BBB and adherent MB231 (red) under no treatment and after the pretreatment of heparinase III (**A**), VEGF (**B**), orosomucoid (**C**), S1P (**D**), and GM 6001 (**E**), and the right plots show the HS intensity of the BBB in the region of interest (ROI) with and without adherent MB231 cells under each case. The image on the upper left corner of (**A**) defines the ROIs with and without adherent MB231 cells (TC). The average HS intensity of the region in between two concentric circles, excluding TCs, (upper left) is that for the ROI with adherent TCs and the average HS intensity of the circled region (lower right) is that for the ROI without adherent TCs. * *p* < 0.05. Values are presented as mean ± SD. *n* ≥ 30 ROIs (at least 15 ROIs with adherent TCs and 15 without adherent TCs) analyzed for each case.

**Figure 7 cells-13-00190-f007:**
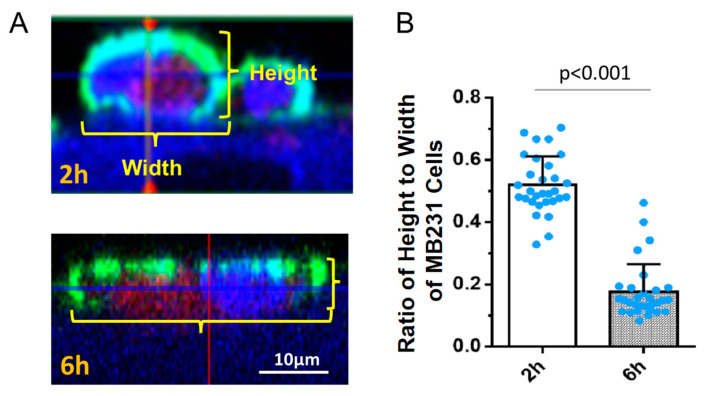
The ratio of height-to-width of MB231 cells as a function of adhesion time on in vitro BBB. (**A**) Confocal images showing the side view of adherent MB231 cells on the BBB formed on a Transwell filter at 2 h (top) and 6 h (bottom). (**B**) Comparison of the ratio of cell height-to-width between 2 h and 6 h adhesion after seeding. Values are presented as mean ± SD. *n* ≥ 30 MB231 cells were analyzed for each time.

## Data Availability

Data are contained within the article.

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
