# Peer review of "Heparan Sulfate Modulation Affects Breast Cancer Cell Adhesion and Transmigration across In Vitro Blood–Brain Barrier"

_cells, 2024, doi:10.3390/cells13020190_

Round 1

Reviewer 1 Report

Comments and Suggestions for Authors

In this study, the authors indicated that reduced/enhanced BBB HS and enhanced/reduced MDA-MB-231 HS increase/decrease its adhesion to and transmigration across the BBB. The study is well designed and of interest. The experiments in vitro have provided sufficient evidence for the conclusions. I would recommend the manuscript to be published. Have the authors validate the conclusion in animal models? As we all know, the animal model is necessary when studying the brain metastasis of breast cancer. Please explain, discuss or add more experiments in vivo.

Reviewer 2 Report

Comments and Suggestions for Authors

The work carried out by the authors of this paper is very well organized and is undoubtedly the result of an enormous research effort of great relevance and impact for the advancement in the study of therapeutics to reduce fatal outcomes due to metastasis.

It is a shame that in its current state it cannot be published as it requires editing all the bar graphs.

In the current state, all the bar graphs in the paper should be edited since the values ​​of the bars are generally misconfigured, the size of them does not make them visible. I suggest that some of them be chosen, the most representative, and presented in a size that facilitates their study, accompanied by a very good explanation that allows for the other cases only to present the results and discuss them.

Reviewer 3 Report

Comments and Suggestions for Authors

Comment 1: In the figures, throughout the results, it's hard to interpret p values representing statistical significance. Maybe it will be better to mention p values corresponding to symbols in figure itself rather than writing these in legend. 

Comment 2:  In Figure1, 2 mention cell lines in results itself. Authors can write them on Images it is easily visible to readers.

Comment 3: Figure 3, it was strange to see only quantification graphs in results, without any representative data. Can authors add either a data image or any cartoon showing how the data was collected and quantified?

Comments on the Quality of English Language

Many sentences are too lengthy that makes it hard to interpret what authors are trying to explain. For example, Introduction line 61-65. Section 3.2 line 292-296. Please reframe and rewrite.

Round 2

Reviewer 2 Report

Comments and Suggestions for Authors

very well done

congrats